# Developing a Tool for Auditing the Quality of Antibiotic Dispensing in Community Pharmacies: A Pilot Study

**DOI:** 10.3390/antibiotics11111529

**Published:** 2022-11-01

**Authors:** Maarten Lambert, Ria Benkő, Athina Chalkidou, Jesper Lykkegaard, Malene Plejdrup Hansen, Carl Llor, Pia Touboul, Indrė Trečiokienė, Maria-Nefeli Karkana, Anna Kowalczyk, Katja Taxis

**Affiliations:** 1Unit of PharmacoTherapy, Epidemiology and Economics, Groningen Research Institute of Pharmacy, University of Groningen, 9713 AV Groningen, The Netherlands; 2Department of Clinical Pharmacy and Albert Szent-Györgyi Medical Center, Central Pharmacy and Emergency Care Department, University of Szeged, 6720 Szeged, Hungary; 3Section of General Practice, Department of Public Health, University of Copenhagen, 1353 Copenhagen, Denmark; 4Audit Project Odense, Research Unit of General Practice, University of Southern Denmark, 5230 Odense, Denmark; 5Center for General Practice, Aalborg University, 9220 Aalborg, Denmark; 6Institut Català de la Salut, Via Roma Health Centre, 08007 Barcelona, Spain; 7Department of Public Health, Nice University Hospital, 06202 Nice, France; 8Institute of Biomedical Sciences, Faculty of Medicine, Pharmacy Center, Vilnius University, 03101 Vilnius, Lithuania; 9Clinic of Social and Family Medicine, Faculty of Medicine, University of Crete, 71003 Heraklion, Crete, Greece; 10Centre for Family and Community Medicine, Faculty of Health Sciences, Medical University of Lodz, 90-153 Lodz, Poland

**Keywords:** community pharmacy practice, dispensing quality, antibiotics, antimicrobial resistance, Audit Project Odense

## Abstract

Background: The European Centre for Disease Prevention and Control describes the community pharmacist as the gatekeeper to the quality of antibiotic use. The pharmacist has the responsibility to guard safe and effective antibiotic use; however, little is known about how this is implemented in practice. Aims: To assess the feasibility of a method to audit the quality of antibiotic dispensing in community pharmacy practice and to explore antibiotic dispensing practices in Greece, Lithuania, Poland, and Spain. Methods: The Audit Project Odense methodology to audit antibiotic dispensing practice was adapted for use in community pharmacy practice. Community pharmacists registered antibiotic dispensing on a specifically developed registration chart and were asked to provide feedback on the registration method. Results: Altogether, twenty pharmacists were recruited in four countries. They registered a total of 409 dispenses of oral antibiotics. Generally, pharmacists were positive about the feasibility of implementing the registration chart in practice. The frequency of checking for allergies, contraindications and interactions differed largely between the four countries. Pharmacists provided little advice to patients. The pharmacists rarely contacted prescribers. Conclusion: This tool seems to make it possible to get a useful picture of antibiotic dispensing patterns in community pharmacies. Dispensing practice does not seem to correspond with EU guidelines according to these preliminary results.

## 1. Introduction

Community pharmacists are in a unique position to positively impact antibiotic use and reduce antimicrobial resistance [1,2]. The European Centre for Disease Prevention and Control (ECDC) has established guidelines for the prudent use of antimicrobials for human consumption, explicitly stating that community pharmacists are gatekeepers to antibiotic use [3]. As gatekeepers, community pharmacists can reduce unnecessary antibiotic use for self-limiting infections and ensure optimal use of antibiotics [3]. In this role, pharmacists act as a source of information for patients and prescribers on the safe, rational, and effective use of antimicrobials [3]. This includes a responsibility to dispense antibiotics based on valid prescriptions which includes checking the rationale for treatment, providing advice, and performing safety checks of contraindications and interactions [3]. Correspondingly, the World Health Organization (WHO) and the International Pharmaceutical Federation have developed guidelines for good pharmacy practice, emphasizing similar responsibilities for community pharmacists [4,5]. Consequently, these organizations advocate the key role that community pharmacists should play in addressing the problem of antimicrobial resistance.

Currently, little is known about dispensing practices for antibiotics in community pharmacies and to what extent pharmacists fulfil the role as gatekeeper to antibiotic use in daily practice. In their systematic review on documenting dispensing practices [6], Cerqueira-Santos et al. stress the need for novel strategies to document the dispensing process to ensure better pharmacy practice with regard to patients and other healthcare professionals. Moreover, as dispensing practices are likely to differ between drug classes, such documenting strategies are preferably specifically adjusted to different drug classes. In order to map antibiotic dispensing practices and gain insight to what extent community pharmacists adhere to current EU guidelines, a specific tool is needed for documenting antibiotic dispensing, as such a tool does not exist yet. Ideally, such a tool must be easy to implement in daily practice and quick to use.

### The Audit Project Odense Methodology

One way to document healthcare practice is through self-registry by healthcare professionals. In general practice for example, the Audit Project Odense (APO) methodology was developed for quality improvement [7] and is used to successfully decrease inappropriate use of antibiotics [8]. The APO method encompasses a bottom-up approach to implement multi-faceted interventions. The core components of this method are two audit registrations [9]. General practitioners register key variables about diagnosis of infectious diseases and prescribing of antibiotics on a pre-specified chart, including patient symptoms, diagnostics, and choice of treatment. In the community pharmacy setting, the APO methodology has not been used previously. Based on the promising results in general practice, applying the APO methodology in the community pharmacy practice setting may be an innovative way to improve antibiotic use through documenting dispensing practices. Therefore, this paper describes the development and pilot testing of an audit chart in the community pharmacy setting. Specifically, the aims of the pilot study are three-fold:To assess the feasibility of registering antibiotic dispensing using the registration chart in the community pharmacy setting;To collect feedback from community pharmacists on the implementation of the APO method;To describe antibiotic dispensing practices in four European countries.

## 2. Results

### 2.1. Feasibility of the APO-Methodology in Community Pharmacy Practice

In total, 20 pharmacies were recruited to participate in the pilot study, five in each of the four countries. One pharmacy in Greece dropped out of the study due to intense workload. All participants (*n* = 19) returned the questionnaire. The participation of pharmacy staff differed between the pharmacies. In 10 pharmacies, both pharmacists and pharmacist technicians participated in the pilot study. In four pharmacies, only one staff member participated; this could either be a pharmacist or a pharmacist technician. In three pharmacies, more than one staff member participated, although not the entire staff. In the remaining two pharmacies, only pharmacists participated. All pharmacies reported that registration of patients took less than one minute per dispensed antibiotic or between one and two minutes, except for one pharmacy that needed two to three minutes per registration. In all countries, pharmacy staff managed to register all patients with a prescription for an oral antibiotic or only missed a couple of dispenses during the study period. Reasons for not registering included high workload or forgetting. Most of the pharmacists found the registration chart, instruction form, and list of antibiotics clear and easy to use.

### 2.2. Antibiotic Dispensing Practice

During the study period, a total of 409 dispenses of antibiotics were registered. Of those antibiotics, 59% were prescribed to female patients, with the average patient age being 43 years (SD = 24 years). The most frequently dispensed antibiotics were amoxicillin and amoxicillin/clavulanic acid, followed by macrolides or clindamycin and cephalosporines, although frequencies differed per country. In total, 77% of the dispenses were registered by pharmacists, 22% by other pharmacy staff, in 2% this was not reported. Nearly half of the dispensed antibiotics were prescribed for acute respiratory tract infections. The indication for the prescribed antibiotic was unknown to the pharmacy staff for 11% of the total number of dispenses. There was contact between the pharmacist and the prescriber for 14 (3%) of the dispenses, which led to changes to the prescription in nine (2%) cases. In Poland, there was no contact with prescribers at all, and in 12% of the dispenses this information was not reported (Table 1).

The frequency of checking for drug-drug interactions, contraindications, and allergies during the dispensing process differed largely between the countries. In total, in 49% of the dispenses none of the three safety checks were performed. However, in 70% of the cases in Lithuania none of the checks were performed, whereas in Greece no checks were performed in 18% of the cases. When looking at the individual safety checks, checking for contraindications was performed the least often (21%) and checking for allergies most often (36%). Only in Spain and Greece were there dispenses for which all three safety checks were performed, in 24% and 22% of registrations, respectively (Table 1).

Overall, in 66% of the dispenses, the pharmacy staff discussed treatment duration with patients. Other general advice that is deemed appropriate to give during dispensing of all antibiotics was given less frequently: information about side effects (21%), informing about risk of AMR (18%), seeking medical help if symptoms worsen (19%), and bringing back leftovers (4%). In 13% of the dispenses, the pharmacist did not provide the patient with any advice (Appendix A).

Treatment duration was unknown for 7% of the dispensed antibiotics. In 70% of the dispenses, the pharmacy staff deemed the prescription appropriate for the specific situation on a clinical basis (e.g., necessity of antibiotic, correct choice of antibiotic, correct dose, correct treatment duration), in 3% the pharmacy staff did not agree, and in 26% the staff reported to not have sufficient information to make this judgement. This information was missing in 1%. In 31 cases, pharmacists judged a prescription as appropriate despite not knowing the indication and/or treatment duration, which was considered as inappropriate agreement (Appendix A). Four antibiotics were dispensed after wait-and-see advice from the prescriber.

### 2.3. Feedback on the Registration Chart

Most feedback was about the domain of advice on the registration chart and instruction form. For example, for “discuss treatment duration” one Spanish pharmacist commented: ‘does this mean to explain and reinforce the importance of not stopping treatment until finishing it, or only explain the duration of treatment?’. Moreover, pharmacists reported they found some advice unnecessary to give while missing other information, although this feedback differed per pharmacy, within and between the countries. Several other topics were suggested to be added to the registration chart, including veterinarian use, probiotics, prophylaxis, injectable antibiotics, metronidazole, treatment preparation, and storage.

Similarly, suggestions were provided for changes to other domains of the registration chart. Pharmacists in Greece described that it was difficult and uncommon to contact prescribers. In Lithuania, pharmacists reported that most of the time it was almost impossible to contact prescribers for clarification or changes to the prescription. Interestingly, the registrations during the pilot study show that in Greece there was contact with the prescriber in 5.5% of the cases and in Lithuania in 8.2% of the cases, whereas in Poland this was 0%. Additionally, some Lithuanian pharmacists mentioned that safety checks for contraindications and interactions were not performed in their pharmacies and patients were usually not informed about side effects from drug use. This aligns with the registered dispenses, as contraindications were only checked in 1.0% of cases, interactions, and allergies in 9.7% and 15.5%, respectively, and information about side effects was provided in 2.9% of the dispenses. Polish pharmacists reported that it was often not possible to give an assessment of the treatment as they did not know the indication for prescriptions and do not have access to patients’ medical history. Despite this, Polish pharmacists only reported an unknown location of infection in 10.8% of the dispenses. Finally, in Spain, the difference between pharmacists and other pharmacy staff was reported by multiple pharmacies. As only pharmacists are allowed to evaluate interactions and contraindications for new patients, it was suggested to exclude technicians from the study. Indeed, the registrations show a difference between pharmacists and non-pharmacists in Spain, as they checked for interactions in 44.0% and 10.5% of the dispenses, respectively, and comparably for contraindications (46.6% vs. 0%) and allergies (62.1% vs. 7.0%).

### 2.4. Revising the Registration Chart for the Main Study

Based on the written feedback that was provided by the participating pharmacy staff and the results obtained during registering the dispensing practice, several changes have been made to the registration chart (Appendix B). Firstly, the total number of answer options was reduced from 46 to 39. This was achieved by changing the location of infection from a choice of infections to a known/unknown question and by removing the domain of delayed prescribing, as this occurred in less than 1% of the dispenses. Metronidazole was added to the domain of antibiotics on request of several pharmacists. Within the domain of advice, some specifications and changes were made. General advice of taking antibiotics with or without food/drinks was changed to more specifically alcohol and dairy products. The advice “do not take shortly before sleeping” and “advice regarding comedication” has been removed from the chart, as the first one was crossed in less than 1% of the dispenses, and for the latter, it is not possible to judge whether this is appropriate due to lack of information of other drug use.

## 3. Discussion

Antibiotic dispensing in community pharmacies is complex and varying practices within countries and across borders exist. This study shows that a simple tool to measure the antibiotic dispensing process can be implemented in community pharmacy practice. When it comes to antibiotic dispensing in community pharmacies, practice does not seem to match EU guidelines. On the one hand, this could mean that proper guidelines should be based on a real-life setting involving practicing pharmacists in establishing such guidelines. On the other hand, registration of dispensing practices using the APO methodology reveals many possibilities for improvement, although the emphasis of such improvements should be dependent on contextual factors within and between countries.

### 3.1. Strengths of the Study

This is the first testing of the APO methodology in community pharmacy practice. The APO methodology has been proven to be effective in general practice over several decades [7,8,10]. During this study, there was close collaboration with the initial developers of the APO methodology in general practice. In addition, the study was conducted in multiple pharmacies in countries with different antibiotic usage and community pharmacy practices. The developed registration chart was easily implemented in all these contexts, suggesting similar high feasibility in a wider range of countries, especially in the EU. Moreover, feedback from the twenty participating pharmacies has been thoroughly reviewed and led to considerable changes to the content of the registration chart, thus improving the adaptation to the field of daily practice. Finally, the research group consisted of a wide range of experts, including experts of the 5 target countries, and practicing community pharmacists.

### 3.2. Limitations of the Study

The complexity of the dispensing process makes it difficult to measure all topics related to it on a registration chart that can be completed within a few minutes. Within that framework, we attempted to include the most relevant parts of the antibiotic dispensing process but had to eliminate or simplify many topics from the registration chart. Several topics have been discussed and considered but not included in the final registration chart. These include registration of multiple other antibiotics and antibiotic classes, symptom assessment of patients without an antibiotic prescription, the patients’ perspective on the dispensing process, patient’s adherence to antibiotic therapy, the use of point-of-care tests, the use of “wait-and-see” prescriptions, and more specific details on safety checks and a wider range of possible appropriate advice. Through this method we developed a registration chart that takes little time to complete. Nevertheless, completing the chart during dispensing will take up additional time of pharmacists, which may mean that implementation might not be possible in all pharmacies for all antibiotic dispenses.

Although it was estimated that the use of antibiotics without a prescription comprised about 7% of total antibiotic use in the EU [11], over-the-counter supply of antibiotics has not been taken up in this pilot study because the extent to which over-the-counter supply occurs differs between the four countries. Moreover, as over-the-counter supply of antibiotics is illegal in the EU, data obtained on this through a self-registry chart might not have been accurate. Other limitations include the limited number of recruited pharmacies in France and the voluntary and non-random participation of participants in the other countries. This does probably mean that the participating pharmacists are more aware of their dispensing practices, they are among the more guideline compliant pharmacists and therefore the results could be biased towards better dispensing practices than what actually happens during daily practice. Moreover, the registration chart was kept consistent for the five target countries, even though pharmacy practice differs between them. This could mean that certain topics on the registration chart may be more relevant in certain countries compared to others. Nevertheless, the final version of the registration chart was developed based on feedback from all countries, where especially those topics that seemed relevant in all contexts were included. Finally, no demographic data were collected for the participating pharmacies, e.g., related to location and size of the pharmacies.

### 3.3. Comparison with Literature

There is only limited literature available on documenting dispensing practice, even more so for antibiotics specifically. Cerqueira Santos et al. [6] reviewed all documentation of dispensing, but included studies mainly focusing on drug-related problems, patient information, and clinical interventions. Although such information seems to be essential for improving pharmacy services, it does not provide information on what exactly happens during the dispensing process. As dispensing practice should differ for different drug classes, specified documentation methods are needed for specific drug classes to ensure obtaining detailed information, which can be used for specific improvements in practice. Studies that focus on antibiotic dispensing have been performed around the globe [12,13,14,15,16,17,18,19,20,21], but mainly aim to identify patterns in dispensing practices, e.g., regarding the type of antibiotic dispensed or over-the-counter dispensing of antibiotics. Such studies seem very relevant to picture general antibiotic use; nonetheless, they might not be as useful in providing specific improvements for community pharmacy practice. As the methodology of this study deviates from earlier research, i.e., the APO methodology has never been used in community pharmacies before, a straightforward comparison with previous literature is difficult to make. Nevertheless, based on the feedback received from the participating pharmacists, it seems that developing and implementing an antibiotic dispensing documentation tool has been feasible and successful. Differences in community pharmacy practice throughout Europe have been reported earlier [22]. Also, with specific regard to the differences in antibiotic use and dispensing practices throughout Europe as shown in this study, similar findings have been published [23] and varied reasons have been identified, including lack of public knowledge and awareness, access to antibiotics without prescription and leftover antibiotics, knowledge and perception of prescribers and dispensers and many others [11,16,24,25,26,27]. However, care must be taken interpreting the data of this pilot study, a study on a larger scale is needed to confirm these.

### 3.4. Meaning of the Study and Future Studies

The ECDC has described a large role for the community pharmacist towards improving the quality of antibiotic use and therewith reducing antimicrobial resistance [3]. Nonetheless, there seems to be a large gap between the role as defined in theory and how community pharmacists fulfil this role in practice. This shows through the few safety checks that are performed and little advice that is given during dispensing and the minimal contact between pharmacists and prescribers. To diminish this gap, strengthen the role of the pharmacist in antibiotic use and hence improve antibiotic dispensing practices, it is essential that two conditions are met. Firstly, a clear picture of current practice is needed to identify problems and possibilities for improvement. The tool we developed in this pilot study might be one method to achieve this, although implementation on a larger scale would provide more convincing evidence. Secondly, pharmacists must be made aware of their role as a gatekeeper as described in the aforementioned guidelines and be given support to change their practice accordingly. It will be important that pharmacists take an active role in this change, looking for multidisciplinary collaboration (e.g., with prescribers) where possible, and striving to improve their practice from within their own profession. Part of the main study of the HAPPY PATIENT project will therefore aim to let community pharmacists gain insight in their daily practice and improve their practice according to EU guidelines using the successfully tested APO methodology [28].

## 4. Materials and Methods

### 4.1. Study Design

This pilot study is part of the Health Alliance for Prudent Prescription and Yield of Antibiotics in a Patient-centered Perspective (HAPPY PATIENT) project. This project aims to further implement the EU AMR guidelines on the prudent use of antimicrobials in humans [3]. The project is supported by the EU Third Health Programme (ID 900024) and focusses on four settings: community pharmacies, general practice, out-of-hour services, and nursing homes. The study protocol has recently been published [28].

### 4.2. Study Setting

Data collection was attempted in 25 community pharmacies, five pharmacies in five different countries with differences in scale and patterns of antibiotic use [29], and spread over different parts of the European Union: France, Greece, Lithuania, Poland, and Spain. Due to difficulties with pharmacist recruitment, only two pharmacists participated in France. To protect the privacy of the French participants, these results were not included in this paper. The local partners in the four countries recruited pharmacists and/or pharmacist technicians working in community pharmacies. Participating staff did not need to speak English as all materials were forward-backward translated into local languages by the local partners. There were no limitations based on pharmacy size, location, or other factors for inclusion in the study.

### 4.3. Development of the Registration Chart

The layout of the registration chart, with multiple variables categorized within overarching domains, was kept consistent with the original audit chart developed for GP practice as earlier published [9,10]. The content of the registration chart was adjusted to suit community pharmacy practice in the target countries. A first draft of the registration chart was developed by ML based on information from two documents: (1) a context analysis of community pharmacy practice in the target countries using a questionnaire which was completed by the local partners of the HAPPY PATIENT project; and (2) the EU AMR guidelines on the prudent use of antimicrobial for humans [3]. Further development of the registration chart, with specification of its domains and variables, was done through online discussion and consensus meetings. The core research group, M.L., R.B., and K.T., determined the focus of the registration chart by selecting appropriate domains and variables, in light of WHO [30,31] and ECDC [3] reports and the official Summary of Product Characteristics (SmPC) texts for antibiotics. For all antibiotics or antibiotic classes included in the registration chart, the SmPC texts were searched for information on recommendations and warnings for use, contraindications, interactions, and precautions. To illustrate, SmPC texts warn for photosensitivity when using tetracyclines, therefore pharmacists are expected to inform patients to be careful with sun- and UV-light when dispensing tetracyclines. Consequently, this advice was included in the registration chart.

The registration chart has been discussed during several meetings with expert groups: the developers of the original GP registration chart, local partners in the target countries, practicing community pharmacists and the complete HAPPY PATIENT project group. The list of antibiotics and antibiotic classes included in the registration chart was composed in collaboration with the local and clinical partners, for consistency throughout the project. The registration chart comprised nine domains with a total of 46 variables related to antibiotic dispensing, and two patient variables—age and sex (Appendix C); it focused on oral antibiotic prescriptions that are dispensed in the community pharmacy. The same chart was used in all four countries.

### 4.4. Data Collection

The registration chart and an instruction document (Appendix D) were distributed among the staff of the participating pharmacies. The instruction document provided general information about the duration of the pilot study, the in- and exclusion criteria for registering, and specific information on the nine domains of the registration chart. Specifically, pharmacy staff was instructed to register all oral antibiotic dispensing inside the pharmacy during 5 working days in October 2021. Antibiotics dispensed outside the pharmacy, e.g., deliveries to patients, were excluded. Any antibiotics prescribed for prophylactic or veterinary use were also excluded from the study. The registration charts were completed on paper, immediately after dispensing. Additionally, a list of antibiotics was provided to support pharmacy staff in assigning specific antibiotics to the appropriate antibiotic class on the registration chart. This list comprised general antibiotics for all countries (Appendix E) and was complemented with country-specific antibiotics and brand names by the local partners in the target countries. Pharmacy staff was instructed to return the charts by postal courier or digital scans to the partners in the target countries. All data were transcribed to IBM^®^ SPSS^®^ and Stata™ files by partners at the Research Unit for General Practice, Institute of Public Health of the University of Southern Denmark.

### 4.5. Questionnaire

To assess the feasibility of implementing the registration chart in practice and to acquire feedback on the registration chart, the pharmacy’s staff was requested to complete a questionnaire following the pilot study (Appendix E). This questionnaire comprised ten questions, on ease of use of the documents (registration chart, instruction form, list of antibiotics), time needed for registrations, possibility to register all antibiotics in the study period, appropriateness of domains and variables, and willingness to participate among the members of the pharmacy’s staff.

### 4.6. Data Analysis

All answers to the questionnaire were translated to English by the partners in the target countries. Due to the small number of participating pharmacies, the received feedback was discussed by the core research group in full. Any unclarities were solved, and suggestions towards increasing the ease of use of the documents or reducing the time needed to complete them were considered if these were relevant in all target countries. Similarly, the content of the registration chart was adjusted based on this questionnaire. To this extent, any topic suggested to include or remove was discussed within the core research group and compared to WHO and ECDC reports and SmPC texts. Topics mentioned by multiple pharmacists were given a higher priority. Any topic was only included if deemed relevant in all four countries and consistent with EU AMR guidelines. Additionally, the data collected with the registration chart was used to further improve its contents.

The data collected with the registration chart were also used to illustrate community pharmacy practice regarding antibiotic dispensing using Stata/MP 16. Data were analyzed descriptively for pharmacies per country and for the countries together. Crosstabs of different combinations of domains were created to analyze combinations of dispensed antibiotics and provided advice. Appropriateness of advice was determined by comparing the collected data to SmPC information for the specific antibiotics. Safety checks of contraindications, interactions, and allergies were deemed to have to be performed for all dispensed antibiotics, as described as the role of the pharmacists in the EU AMR guidelines [3].

## 5. Conclusions

The registration chart based on the APO methodology appears to be a feasible way to obtain detailed data on the antibiotic dispensing practices in community pharmacies. Pharmacists from different countries have been able to implement the registration chart in their daily practice. Although the complex process of antibiotic dispensing cannot be documented entirely within a few minutes, this tool does make it possible to obtain useful information about antibiotic dispensing. Nevertheless, the effectiveness of this tool is not solely based on its design; it will substantially depend on the implementation of interventions that result from using the tool in practice. This pilot study indicates the presence of considerable inconsistencies between the EU guidelines on dispensing and the everyday practices in the pharmacies.

## Figures and Tables

**Table 1 antibiotics-11-01529-t001:** Characteristics of registered dispenses.

	Greece	Lithuania	Poland	Spain	Total	Total (%)	Missing
Dispenses registered	55 (13.4%)	103 (25.2%)	74 (18.1%)	177 (43.3%)	409	100	
Sex							0 (0%)
Female	32	69	42	97	240	58.7	
Male	23	34	32	80	169	41.3	
Education							7 (1.7%)
Pharmacist	38	92	68	116	314	76.8	
Not pharmacist	17	8	6	57	88	21.5	
Antibiotics dispensed							2 (0.5%)
Penicillin V or pivmecillinam	0	0	2	2	4	1.0	
Amoxicillin	7	20	6	36	69	16.9	
Amoxicillin + clavulanic acid	17	25	13	31	86	21.0	
Fosfomycin	0	1	1	29	31	7.6	
Nitrofurantoin	0	10	1	2	13	3.2	
Trimethoprim +/− Sulphonamides	0	4	2	1	7	1.7	
Macrolides or clindamycin	1	9	26	31	67	16.4	
Tetracyclines	2	9	4	2	17	4.2	
Cephalosporins	11	12	10	16	49	12.0	
Quinolones	12	2	5	19	38	9.3	
Other	5	9	4	8	26	6.4	
Focus of infection							1 (0.2%)
Respiratory tract	28	42	52	80	202	49.4	
Urinary tract	7	16	6	45	74	18.1	
Gastrointestinal	6	4	1	11	22	5.4	
Skin	2	1	5	11	19	4.7	
Gynaecological	1	1	1	0	3	0.7	
Other	9	10	1	23	43	10.5	
Unknown	2	29	8	6	45	11.0	
Safety checks performed							
Interactions	25	10	23	58	116	28.4	
Contraindications	20	1	8	55	84	20.5	
Allergies	38	16	15	78	147	35.9	
None of the above	10	72	38	82	202	49.4	
All safety checks performed	12	0	0	42	54	13.2	
Prescriber contact							49 (12.0%)
Yes, and changes to prescription	0	8	0	1	9	2.2	
Yes, no changes to prescription	2	0	0	3	5	1.2	
No contact with prescriber	34	89	74	149	346	84.6	
Pharmacy judgement of prescription							2 (0.5%)
Agree with prescription	39	86	51	110	286	69.9	
Do not agree with prescription	5	3	0	6	14	3.4	
Insufficient information to decide	10	13	23	61	107	26.2	

## Data Availability

The data presented in this study are available on request from the corresponding author. The data are not publicly available due to the small number of participating pharmacists and their privacy.

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
