# Peer review of "Developing a Tool for Auditing the Quality of Antibiotic Dispensing in Community Pharmacies: A Pilot Study"

_antibiotics, 2022, doi:10.3390/antibiotics11111529_

Round 1

Reviewer 1 Report

Manuscript is a good contribution to the readers as it contains novel information and new data. Very well presented.

Summary

Manuscript is original with novelty. Readers will find new information’s and importance of the scientific data presented in the article. Manuscript is presented in a good way. All things have been described in appropriate way. After a few corrections manuscript may be accepted for publication

Specific Comments

Abstract, introduction, methodology, results, and discussion have been presented and written very well. Only order is required to be changed for placement of result should be changed i.e., after the methodology section.

In line # 66, placement of “yet” is required to be adjusted

In line # 94 & 95, rephrase the sentence to make it more clear

Rest of the manuscript needs no corrections in my opinion.

Reviewer 2 Report

The manuscript is very original and this is a good idea for the implementation of the evaluation of the rational use of antibiotics.

There are, however, some methodological limitations that the authors mentioned but perhaps not very clearly referred to. It is important how the pharmacies were selected for the study (on the basis of which criteria, e.g. annual turnover of profits, size of the city in which the pharmacy was located, so that the data could be comparable). I also did not find information on how long the data collection lasted and in what period (e.g. during the fall-winter season, when oral antibiotics are more often prescribed).

Moreover, the validity of using such gudeline may be questionable if there is no contact between the prescribers and the pharmacist.

In general, the idea is very good, only the possibility of its implementation may be problematic, because the time allocated to one patient is still longer than usual, so not all pharmacists will follow it

Reviewer 3 Report

The current study addresses a very important and interesting topic which is the assessment of adequate antibiotic dispensing in community pharmacies. Although it includes data from 4 different countries (Greece, Lithuania, Poland and Spain), which is a plus, only 20 pharmacists were recruited. This is a very small sample size, which does not allow to draw generalized conclusions, even as a “pilot study”. The study should be extended to more pharmacies/pharmacists, and if possible to more different countries. Also the grand majority of the manuscript consists of Appendix data. For all these reasons, I do not advise the acceptance of this manuscript for publication in Antibiotics.

Round 2

Reviewer 3 Report

The changes that the authors have made have improved the manuscript substantially.